# CHIMLE: Conditional Hierarchical IMLE for Multimodal Conditional Image Synthesis

**Shichong Peng[1], Alireza Moazeni[1], Ke Li[1,2]**
[1]APEX Lab
School of Computing Science      [2]Google
Simon Fraser University
{shichong_peng,seyed_alireza_moazenipourasil,keli}@sfu.ca

## Abstract

A persistent challenge in conditional image synthesis has been to generate diverse output images from the same input image despite only one output image being observed per input image. GAN-based methods are prone to mode collapse, which leads to low diversity. To get around this, we leverage Implicit Maximum Likelihood Estimation (IMLE) which can overcome mode collapse fundamentally. IMLE uses the same generator as GANs but trains it with a different, non-adversarial objective which ensures each observed image has a generated sample nearby. Unfortunately, to generate high-fidelity images, prior IMLE-based methods require a large number of samples, which is expensive. In this paper, we propose a new method to get around this limitation, which we dub Conditional Hierarchical IMLE (CHIMLE), which can generate high-fidelity images without requiring many samples. We show CHIMLE significantly outperforms the prior best IMLE, GAN and diffusion-based methods in terms of image fidelity and mode coverage across four tasks, namely night-to-day, $16\times$ single image super-resolution, image colourization and image decompression. Quantitatively, our method improves Fréchet Inception Distance (FID) by 36.9% on average compared to the prior best IMLE-based method, and by 27.5% on average compared to the best non-IMLE-based general-purpose methods. More results and code are available on the project website at https://niopeng.github.io/CHIMLE/.

## 1 Introduction

Impressive advances in image synthesis have been made by generative models [9, 40, 83, 15, 78, 31, 38]. Generative models are probabilistic models; in the context of image synthesis, they aim to learn the probability distribution over natural images from examples of natural images. Whereas unconditional generative models learn the *marginal* distribution over images, conditional generative models learn the *conditional* distribution over images given a conditioning input, such as a class label, textual description or image.

Conditional generative modelling is more challenging than unconditional generative modelling, due to the need to ensure consistency between the conditioning input and the generated output while still ensuring image fidelity and diversity. As the conditioning input becomes more specific, there are fewer images in the training set that are consistent with the conditioning input, resulting in weaker supervision. So, in the order of decreasing supervision are class, text and image-conditional generative models – in class conditioning, many images correspond to each class; in text conditioning, multiple images can at times share the same textual description; in image conditioning (e.g., a grayscale image as the input and a colour image as the output), typically only one output image that corresponds to each input image is observed (e.g., only one way to colour a grayscale image is observed). We

36th Conference on Neural Information Processing Systems (NeurIPS 2022).

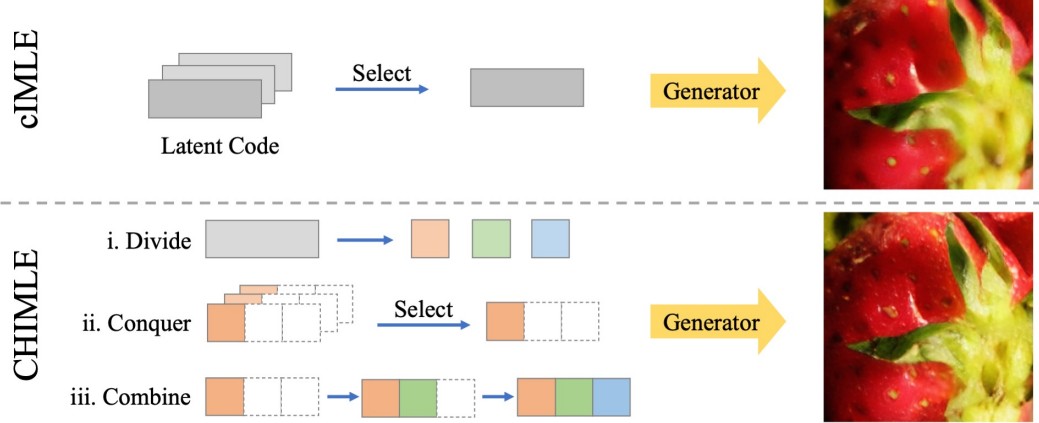

Figure 1: Overview of the prior best IMLE-based method, cIMLE [53] (top row) and the proposed method, CHIMLE (bottom row). Whereas cIMLE samples many latent codes and selects the one that produces the best result, CHIMLE divides the latent code by dimension into components, selects the best sample for each component and iteratively combines the selected components to form the overall latent code. As shown, the sample from CHIMLE contains more fine-grained details than the one from cIMLE.

will focus on the most challenging of these settings, namely image conditioning. Regardless of the supervision, the goal of conditional generative models is to learn the full distribution over all possible images consistent with the input.

Attaining this goal is challenging, especially when the supervision is weak, and so popular existing methods such as variational autoencoders (VAEs) [45, 70] and generative adversarial nets (GANs) [26, 27] focus on generating a point estimate of the conditional distribution, such as the mean (VAE) or an arbitrary mode (GAN). Such methods are known as *unimodal methods*, whereas methods that aim to generate samples from the full distribution are known as *multimodal methods*, since they enable samples to be drawn from as many modes as there exist.

Extending popular unconditional generative models such as VAEs, GANs and diffusion models to the image-conditional multimodal setting has proven to be challenging. VAEs suffer from posterior collapse [18, 57] where the encoder ignores the conditional input, leading the decoder to produce the conditional mean, which typically corresponds to a blurry image. GANs suffer from mode collapse [37, 105], where the generator tends to ignore the latent noise, leading the generator to produce the same mode of the conditional distribution for the same input regardless of the latent noise. As a result, other equally valid modes (e.g., other possible ways to colour a grayscale image) cannot be generated. Recent and concurrent work [42, 6] extends diffusion models to this setting, but they can produce both too little and too much diversity by creating spurious modes and dropping modes. We dub this problem *mode forcing* and include an analysis of its cause in the appendix.

A recent method [53] takes a different approach by extending an alternative generative modelling technique known as Implicit Maximum Likelihood Estimation (IMLE) [52] to the conditional synthesis setting. At a high level, IMLE uses a generator like GANs, but instead of using an adversarial objective which makes each generated sample similar to some *observed image*, IMLE uses a different objective that ensures each observed image has some similar *generated samples*. Hence, the generator can cover the whole data distribution without dropping modes. We refer interested readers to [53, 52] for more details on the algorithm.

However, as shown in the top row of Figure 1, the output image from cIMLE [53] lacks fine details. For cIMLE to generate high quality samples, there must a sample out of $m$ samples generated during training that is similar to the observed image. This necessitates a large $m$; unfortunately, generating a large number of samples is computationally expensive. So we are faced with a tradeoff: do improvements in sample quality have to come at the expense of sample efficiency?

In this paper, we show that there is a way around this conundrum. The idea is to generate samples in a clever way such that the best sample is about as similar to the observed image as if a large number of samples had been generated, *without* actually generating that many samples. We propose three ideas: partitioning of the latent code, partial evaluation of latent code components and iterative construction of latent code as shown in the bottom row of Figure 1. These three ideas give rise to a novel method known as Conditional Hierarchical IMLE (CHIMLE). We demonstrate that CHIMLE significantly outperforms the prior best IMLE-based method [53] in terms of both fidelity and diversity across four challenging tasks, namely night-to-day, $16\times$ single image super-resolution, image colourization and image decompression. Moreover, we show that CHIMLE achieves the state-of-the-art results in image fidelity and mode coverage compared to leading general-purpose multimodal and task-specific methods, including both GAN-based and diffusion-based methods.

## 2 Method

### 2.1 Preliminaries: Conditional Implicit Maximum Likelihood Estimation (cIMLE)

In ordinary unimodal synthesis, the model is a function $f_\theta$ parameterized by $\theta$ that maps the input to the generated output. To support multimodal synthesis, one can add a latent random variable as an input, so now $f_\theta$ takes in both the input $\mathbf{x}$ and a latent code $\mathbf{z}$ drawn from a standard Gaussian $\mathcal{N}(0, \mathbf{I})$ and produces an image $\widehat{\mathbf{y}}$ as output. To train such a network, we can use a conditional GAN (cGAN), which adds a discriminator network that tries to tell apart the observed image $\mathbf{y}$ and the generated output $\widehat{\mathbf{y}}$. The generator is trained to make its output $\widehat{\mathbf{y}}$ seem as real as possible to the discriminator. Unfortunately, after training, $f_\theta(\mathbf{x}, \mathbf{z})$ often produces the same output for all values of $\mathbf{z}$ because of mode collapse. Intuitively, this happens because making $\widehat{\mathbf{y}}$ as real as possible would push it towards the observed image $\mathbf{y}$, so the generator tries to make its output similar to the observed image $\mathbf{y}$ for all values of $\mathbf{z}$. As a result, naïvely applying cGANs to the problem of *multimodal* synthesis is difficult.

Conditional IMLE (cIMLE) [53] proposes an alternative technique for training the generator network $f_\theta$. Rather than trying to make *all* outputs generated from different values of $\mathbf{z}$ similar to the observed image $\mathbf{y}$, it only tries to make *some* of them similar to the observed image $\mathbf{y}$. The generator is therefore only encouraged to map one value of $\mathbf{z}$ to the observed image $\mathbf{y}$, allowing other values of $\mathbf{z}$ map to *other* reasonable outputs that are not observed. This makes it possible to perform *multimodal* synthesis. Also, unlike cGANs, cIMLE does not use a discriminator and therefore obviates adversarial training, making training more stable. The cIMLE training objective is:

$$\min_\theta \mathbb{E}_{\mathbf{z}^{(1,1)},\ldots,\mathbf{z}^{(n,m)} \sim \mathcal{N}(0,\mathbf{I})} \left[ \sum_{i=1}^n \min_{j \in \{1,\ldots,m\}} d(f_\theta(\mathbf{x}^{(i)}, \mathbf{z}^{(i,j)}), \mathbf{y}^{(i)}) \right], \tag{1}$$

where $d(\cdot, \cdot)$ is a distance metric, $m$ is a hyperparameter, and $\mathbf{x}^{(i)}$ and $\mathbf{y}^{(i)}$ are the $i^{\text{th}}$ input and observed image in the dataset. During each iteration of the training loop, a *pool* of $m$ samples are generated from $f_\theta$ for each input $\mathbf{x}^{(i)}$. The cIMLE algorithm then selects the *closest* sample for each observed image $\mathbf{y}^{(i)}$ to optimizes $\theta$.

### 2.2 Conditional Hierarchical IMLE (CHIMLE)

In order to generate high-quality images using cIMLE, we need some of the $m$ generated samples to be similar to the observed image $\mathbf{y}$ during training. Achieving this requires a large $m$, but generating samples is expensive. This forces a tradeoff between the number of samples and the mode modelling accuracy, which is less than ideal.

The key observation is that the gradient of the loss depends solely on the *closest* sample for each observed image, as shown in Eqn. 1 – all other samples that are not selected do not contribute. Therefore, if we could efficiently search for a latent code $\mathbf{z}$ that would produce a sample of a comparable degree of similarity to $\mathbf{y}$ as one that would be selected from a larger pool $\mathbf{z}^*$, we could achieve high sample quality without actually generating that many samples.

To solve search problems, a time-tested paradigm in computer science is divide-and-conquer, which entails *dividing* the problem into simpler sub-problems, *conquering* each sub-problem and *combining* the solutions from the solved sub-problem. In the context of searching for a latent code $\mathbf{z}$ that

simulates the quality of $\mathbf{z}^*$, we propose dividing the latent code into components, evaluating each latent code component and combining the latent code components iteratively.

In the ensuing discussion, we will consider a common paradigm of image synthesis models [68, 37, 13, 39], where the model processes the image at $L$ different scales. We can feed different dimensions of the latent code as inputs to different layers operating at different resolutions, thereby controlling the variations of content at different resolutions.

**Division of Latent Code**    To reduce the difficulty of the original problem, we can divide it into sub-problems. Each of these sub-problems should be a simpler instance of the original problem. We propose dividing the dimensions of the latent code $\mathbf{z} \in \mathbb{R}^n$ into $L$ different components $S = \{\mathbf{z}_i\}_{i=1}^L$, where $\mathbf{z}_i \in \mathbb{R}^{k_i}$ and $\sum_{i=1}^L k_i = n$, which amounts to dividing the latent space into mutually orthogonal subspaces. Each component $\mathbf{z}_i$ corresponds to the dimensions that serve as input to layers operating at a particular resolution.

**Partial Evaluation of Latent Code Components**    To determine how promising a latent code component is, we need a way to evaluate a partially constructed latent code, with only some components, indexed by $S_r \subset S$, being realized with concrete values. We propose using a partial output $\tilde{\mathbf{y}}_p$ that depends only on the currently constructed components $\mathbf{z}_{S_r}$, and evaluating $\tilde{\mathbf{y}}_p$ against a part of the observed image $\mathbf{y}_p$ that only contains details at resolutions modelled by $\mathbf{z}_{S_r}$. So instead of computing $d(\tilde{\mathbf{y}}, \mathbf{y})$, we compute $d(\tilde{\mathbf{y}}_p, \mathbf{y}_p)$. By doing so, we can evaluate the quality of $\mathbf{z}_{S_r}$ more accurately, since $\tilde{\mathbf{y}}_p$ does not depend on the unconstructed components $\mathbf{z}_{S \setminus S_r}$. To realize this, we add output heads at the $L$ different resolutions that the model operates on to produce the partial outputs $\tilde{\mathbf{y}}_p$ and downsample the observed image $\mathbf{y}$ to the corresponding resolutions to produce the partial observed images $\mathbf{y}_p$. The values of the partially realized components that result in the greatest similarity to $\mathbf{y}_p$ is the final selected component, i.e., $\mathbf{z}_p^* = \arg\min d(\tilde{\mathbf{y}}_p, \mathbf{y}_p)$.

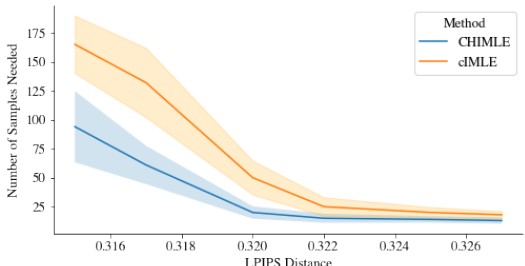

---

**Algorithm 1** Conditional Hierarchical IMLE Algorithm for a Single Data Point

---

**Require:** One input image at $L$ increasing resolutions $\{\mathbf{x}_l\}_{l=1}^L$, the set of partial observed images $\{\mathbf{y}_l\}_{l=1}^L$ at corresponding resolutions and the generator $f_\theta(\cdot, \cdot)$ that produces $L$ outputs at corresponding resolutions

**for** $l = 1$ **to** $L$ **do**

    Randomly generate $m$ i.i.d. latent codes for the $l^{\text{th}}$ component $\mathbf{z}_l^{(1)}, \ldots, \mathbf{z}_l^{(m)}$

    $\tilde{\mathbf{y}}_l^{(j)} \leftarrow l^{\text{th}}$ output from $f_\theta((\mathbf{x}_1, \ldots, \mathbf{x}_l), (\mathbf{z}_1^*, \ldots, \mathbf{z}_{l-1}^*, \mathbf{z}_l^{(j)})) \; \forall j \in \{1, \ldots, m\}$

    $\mathbf{z}_l^* \leftarrow \arg\min_j d(\mathbf{y}_l, \tilde{\mathbf{y}}_l^{(j)}) \; \forall j \in \{1, \ldots, m\}$

**end for**

**return** $\{\mathbf{z}_l^*\}_{l=1}^L$

---

Figure 2: Generated sample quality comparison between the proposed Conditional Hierarchical IMLE (CHIMLE) and cIMLE [53]. The plot shows the number of generated samples needed to reach the same distance (as measured by LPIPS) to the observed image with CHIMLE or cIMLE. A smaller number of generated samples means better efficiency in finding a sample that is close to the observed image. The result is averaged over ten independent runs. As shown, CHIMLE consistently outperforms cIMLE by requiring significantly fewer samples. Also, the more stringent the required LPIPS distance gets, the greater the disparity between the number of samples needed by cIMLE vs. CHIMLE.

**Iterative Combination of Latent Code Components**    To maximize efficiency, the solution to one sub-problem should reveal some information on which other sub-problems to solve. So, we need to decide on which sub-problems to solve first and how to use their solutions to decide on which sub-problem to solve next. We leverage the fact that solving for the appearance at lower resolutions (e.g., finding out that the coarse structure of an apple is red) can inform on the probable appearances

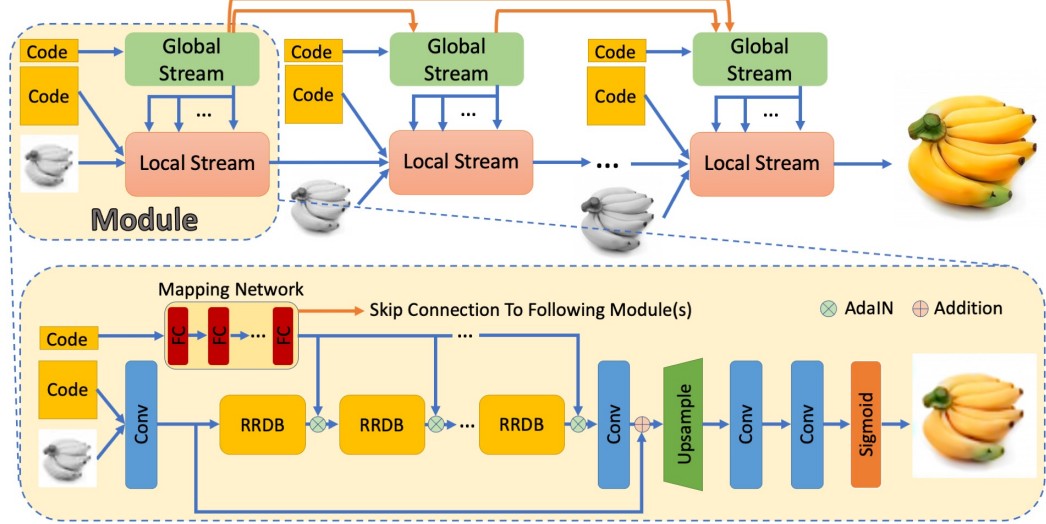

Figure 3: Our model consists of multiple modules, each of which operates on $2\times$ the resolution of the previous one. Each module contains a global stream and a local stream. The local stream consists of a sequence of residual-in-residual dense blocks (RRDB). The global stream consists of a mapping network which applies adaptive instance normalization (AdaIN) to the output of each RRDB block.

at higher resolutions (e.g., the fine details of the apple are likely red as well). We propose starting the construction of the latent code from the component operating at the lowest resolution to the highest, i.e. from $z_1$ to $z_L$, where $z_i$'s are assumed to be ordered by resolution in increasing order. At a given resolution $l \in \{1 \dots L\}$, we set the components at all lower resolutions to the values selected in previous steps $z_1^*, \dots, z_{l-1}^*$ to construct $z_{S_l} = (z_1^*, \dots, z_{l-1}^*, z_l)$, and evaluate the quality of $z_{S_l}$ using the partial evaluation method proposed above.

Putting everything together, we obtain the Conditional Hierarchical IMLE (CHIMLE) algorithm, which is detailed in Alg. 1 for the special case of one input image. We validate the effectiveness of CHIMLE by comparing the number of generated samples CHIMLE needs to obtain the same level of similarity to the observed image as cIMLE. As shown in Figure 2, CHIMLE consistently requires fewer samples than cIMLE, demonstrating a significant improvement in sample efficiency.

### 2.3 Model Architecture

As shown in Figure 3, our architecture contains a sequence of modules, each of which handles an input image of a particular resolution and outputs an image whose resolution is doubled along each side. We downsample the input image repeatedly by half to obtain input images at different resolutions, which are fed into different modules. As mentioned in Sect. 2.2, we divide the dimensions of the latent code into components and feed each to a different module. Note that this architecture generalizes to varying levels of output resolution, since we can simply add more modules for high-resolution outputs. We add intermediate output heads to each module and add supervision to the output of each module. We choose LPIPS [101] as the distance metric used in the IMLE objective function.

For the design of each module, we follow the paradigm of prior work on modelling global and local image features [59, 10, 84, 58, 74] and design a module that comprises of two branches, a local processing branch and a global processing branch. The local processing branch takes in the conditional image input, a latent code with the matching resolution as the image and the output from the previous module if applicable. The global processing branch takes in a fixed-sized latent code input and produces scaling factors and offsets for the different channels of the intermediate features in the local processing branch. For more implementation details, please refer to the appendix. Due to the hierarchical nature of the architecture and its role as the implicit model in IMLE (also known as the "generator" in GAN parlance), we dub this architecture the Tower Implicit Model (TIM).

## 3 Experiments

**Baselines**  To validate our main contribution of this paper, namely improving the output fidelity of IMLE-based methods, we compare to the leading IMLE-based method, cIMLE [53]. As a

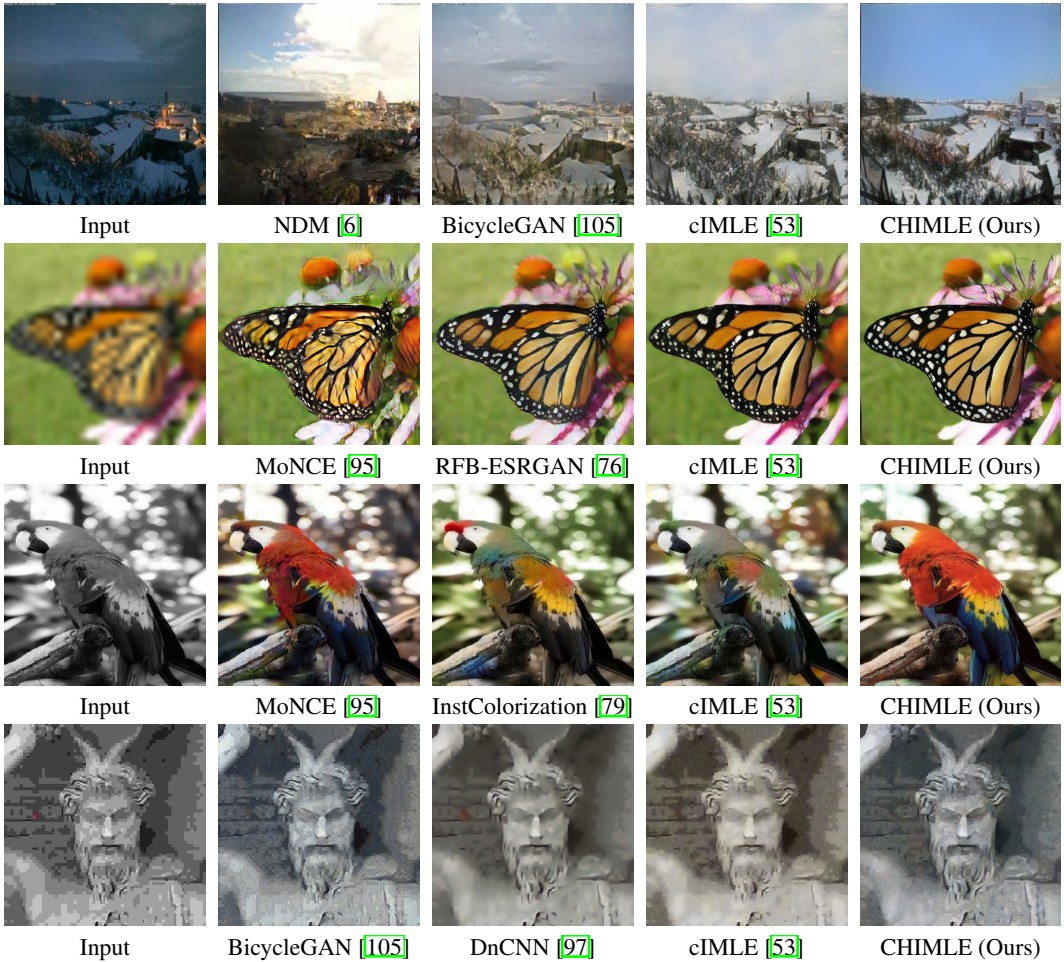

| Input | NDM [6] | BicycleGAN [105] | cIMLE [53] | CHIMLE (Ours) |
| Input | MoNCE [95] | RFB-ESRGAN [76] | cIMLE [53] | CHIMLE (Ours) |
| Input | MoNCE [95] | InstColorization [79] | cIMLE [53] | CHIMLE (Ours) |
| Input | BicycleGAN [105] | DnCNN [97] | cIMLE [53] | CHIMLE (Ours) |

Figure 4: Qualitative comparison of our method (CHIMLE) and selected baselines on night-to-day (1st row), 16× super-resolution (2nd row), image colourisation (3rd row) and image decompression (4th row). In each row, we show the results from the best performing general-purpose baseline, a strong task-specific baseline (if applicable), the best prior IMLE-based method cIMLE [53] and CHIMLE. Better viewed zoomed-in. As shown, CHIMLE correctly preserves the details of the input while generating fine-grained details in the output, which compares favourably to the baselines.

secondary comparison, we demonstrate potential impact in a broader context and compare our method to the leading multimodal general-purpose conditional image synthesis methods according to a recent survey [65], including four GAN-based methods, such as BicyleGAN [105], MSGAN [60], DivCo [56] and MoNCE [95], and two diffusion-based methods, such as DDRM [42] and NDM [6]. As a tertiary comparison, we also compare our method to popular task-specific methods based on leaderboard rankings in challenges and recent survey papers [96, 3].

**Tasks**  We apply our method to four different conditional image synthesis tasks, namely night-to-day, 16× single image super-resolution, image colourisation and image decompression. We pick night-to-day because it is a classic task in the multimodal general-purpose conditional image synthesis literature [37, 105]. We pick the remaining tasks because they are more challenging than typical tasks considered in the literature as evidenced by the availability of task-specific methods tailored to each.

**Training Details**  We use a four-module TIM model (described in Sect. 2.3) for all tasks. The input to each module is downsampled from the full resolution input to resolution the module operates on. We trained all models for 150 epochs with a mini-batch size of 1 using the Adam optimizer [44] on an NVIDIA V100 GPU. Details for datasets are included in the appendix.

**Evaluation Metrics**  We evaluate the visual fidelity of generated output images using the Fréchet Inception Distance (FID) [30] and Kernel Inception Distance (KID)  [7]. We also measure the diversity of the generated output images since it is a crucial objective of *multimodal* conditional image

synthesis. One metric to measure diversity is the LPIPS diversity score (not to be confused with the LPIPS distance metric) [105], which is the average LPIPS distance between different output samples for the same input. Unfortunately it has limitations, because it may favour a set of samples where some are of poor fidelity as shown in Figure 5. To get around this, we use faithfulness-weighted variance (FwV) [53] instead, which is the average LPIPS distance $d_{\text{LPIPS}}$ between the output samples and the mean, weighted by the consistency with the original image measured by a Gaussian kernel [1]:

$$\mathcal{M} = \sum_i \sum_j w_i d_{\text{LPIPS}}(\tilde{\mathbf{y}}^{(i,j)}, \bar{\mathbf{y}}^{(j)}), \text{where } w_i = \exp\left(\frac{-d_{\text{LPIPS}}(\tilde{\mathbf{y}}^{(i,j)}, \mathbf{y}^{(j)})}{2\sigma^2}\right) \tag{2}$$

where $\tilde{\mathbf{y}}^{(i,j)}$ is the $i^{\text{th}}$ generated sample from the $j^{\text{th}}$ input. $\bar{\mathbf{y}}^{(j)}$ is the mean of all generated samples from the $j^{\text{th}}$ input. $\mathbf{y}^{(j)}$ is the $j^{\text{th}}$ observed image. The kernel bandwidth parameter $\sigma$ trades off the importance of consistency vs. diversity.

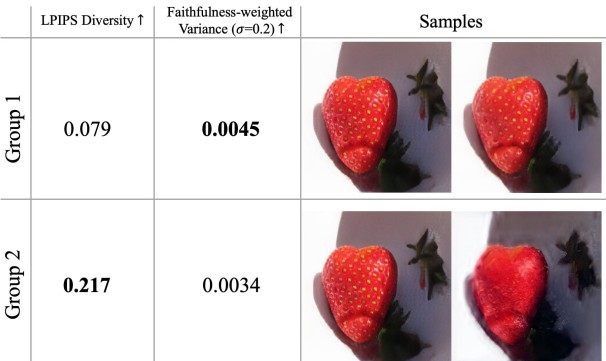

| | LPIPS Diversity ↑ | Faithfulness-weighted Variance ($\sigma$=0.2) ↑ | Samples |
|---|---|---|---|
| Group 1 | 0.079 | **0.0045** | |
| Group 2 | **0.217** | 0.0034 | |

Figure 5: Comparisons of LPIPS diversity score and faithfulness-weighted variance on two groups of sample images. Although Group 2 achieves a better LPIPS diversity score (shown on the left), some of the samples have poor visual fidelity. Group 1 contains samples that are both diverse (zoom in for difference in the seeds on the surface of the strawberry) and of high visual fidelity, which is preferred by the faithfulness-weighted variance (shown on the left).

We measure how well the modes in the model distribution and the true distribution match along two axes, mode accuracy (precision) and mode coverage (recall). We measure the former with the $F_{1/8}$ score component of the PRD metric [71], which is heavily weighted towards precision, and the latter with the $F_8$ score component of the PRD metric [71], which is heavily weighted towards recall, and the inference-via-optimization (IvO) metric [61], which measures the final mean squared error (MSE) between the the observed image and the best output image that can be generated (i.e., where the latent code is optimized w.r.t. MSE while keeping model weights fixed). In the conditional image synthesis setting, IvO is a more sensitive metric than the $F_8$ score in PRD, since it would detect cases where the output samples for a given input are not diverse but the output samples for different inputs are diverse in aggregate. In other words, IvO measures mode coverage in the conditional distribution, whereas the $F_8$ score measures mode coverage in the marginal distribution. In light of the large number of noise samples in NDM (2 for each of the 1000 diffusion steps) which all need to be optimized in calculating IvO, we omit it due to the lack of sufficient computational resources.

### 3.1 Quantitative Results

We compare the perceptual quality and output diversity in Table 1. As shown, CHIMLE outperforms best prior IMLE-based method, cIMLE, general-purpose conditional image synthesis methods and task-specific methods in terms of FID and KID across all tasks. CHIMLE also outperforms all multimodal baselines in terms of FwV across all tasks. These results show that CHIMLE can produce more realistic and diverse images than the baselines, thereby setting a new state-of-the-art.

We compare the mode accuracy and coverage in Table 2. As shown, CHIMLE outperforms or achieves comparable performance to cIMLE and other multimodal general-purpose baselines in

---

[1]($\exp\left(-d_{\text{LPIPS}}(\cdot, \cdot)/2\sigma^2\right)$ is considered a Gaussian kernel rather than a Laplacian kernel because LPIPS distance is defined as a *squared* Euclidean distance in feature space.)

| | Night-to-day | | | Super-Resolution (SR) | | | Colourization (Col) | | | Image Decompression (DC) | | |
|---|---|---|---|---|---|---|---|---|---|---|---|---|
| | *FID* ↓ | *KID* ↓ | *FwV* ↑ | *FID* ↓ | *KID* ↓ | *FwV* ↑ | *FID* ↓ | *KID* ↓ | *FwV* ↑ | *FID* ↓ | *KID* ↓ | *FwV* ↑ |
| General-Purpose Methods: | | | | | | | | | | | | |
| *GAN-based:* | | | | | | | | | | | | |
| *BicyleGAN [105]* | 179.08 | 132.35 | 0.23 | 67.17 | 53.28 | 0.84 | 53.33 | 34.44 | 7.44 | 87.35 | 20.24 | 0.29 |
| *MSGAN [60]* | 213.81 | 176.55 | 0.18 | 83.65 | 67.47 | 0.76 | 53.53 | 31.12 | 10.23 | 100.58 | 27.75 | 0.31 |
| *DivCo [56]* | 179.85 | 136.47 | 0.22 | 66.97 | 54.86 | 0.65 | 53.82 | 34.44 | 12.49 | 92.76 | 24.44 | 0.25 |
| *MoNCE [95]* | 232.10 | 217.79 | 0.18 | 47.97 | 35.31 | 0.68 | 27.67 | 9.55 | 18.30 | 113.21 | 27.56 | 1.12 |
| *Diffusion-based:* | | | | | | | | | | | | |
| *DDRM [42]* | 324.01 | 258.69 | 0.04 | 178.62 | 170.72 | 0.80 | 138.88 | 121.77 | 3.06 | 157.00 | 61.08 | 0.68 |
| *NDM [6]* | 167.50 | 132.93 | 0.28 | 118.30 | 106.51 | 1.84 | 43.98 | 16.21 | 14.39 | 136.32 | 52.64 | 0.67 |
| Task-Specific Methods: | | | | | | | | | | | | |
| *RFB-ESRGAN (SR) [76]* | – | – | – | 19.90 | 8.01 | – | – | – | – | – | – | – |
| *InstColorization (Col) [79]* | – | – | – | – | – | – | 26.37 | 8.52 | – | – | – | – |
| *DnCNN (DC) [97]* | – | – | – | – | – | – | – | – | – | 109.38 | 48.05 | – |
| IMLE-based Methods: | | | | | | | | | | | | |
| *cIMLE [53]* | 166.28 | 123.15 | 0.47 | 28.42 | 14.75 | 5.22 | 63.00 | 49.30 | 14.65 | 101.66 | 30.11 | 3.14 |
| *CHIMLE (Ours)* | **141.44** | **90.10** | **0.51** | **16.01** | **4.54** | **5.61** | **24.33** | **7.73** | **27.11** | **73.69** | **12.12** | **3.80** |

Table 1: Comparison of fidelity of generated images, measured by the Fréchet Inception Distance (FID) and Kernel Inception Distance (KID), and diversity, measured by faithfulness-weighted variance (FwV). We show the results by our method (CHIMLE) and the leading IMLE-based, task-specific baselines and general-purpose conditional image synthesis baselines. Lower values of FID/KID are better and a higher value of faithfulness-weighted variance shows more variation in the generated samples that are faithful to the observed image. For FwV, we show the results for the bandwidth parameter $\sigma = 0.2$ as it achieves a good balance between consistency and diversity. The KID and the FwV are shown on the scale of $10^{-3}$. "−" indicates metric not applicable. We compare favourably relative to the baselines.

terms of PRD scores. In addition, CHIMLE significantly outperforms cIMLE and other multimodal general-purpose baselines in terms of IvO. These results show that CHIMLE achieves better mode accuracy and coverage of the true distribution compared to the baselines.

## 3.2 Qualitative Results

We show the qualitative comparison of our method (CHIMLE) and selected baselines on various tasks in Figure 4. As shown, CHIMLE generates high-quality results, like the high contrasts in the building and the bushes in night-to-day (1st row), the sharp appearance of the wing pattern of the butterfly wing and the flowers in the background (2nd row), the vibrant yet detailed colouring of the parrot (3rd row) and the high-frequency details in the sculpture (4th row). For all methods, we show videos of different samples generated from the same input on the project website and in the supplementary materials. Due to mode collapse, GAN-based baselines incorporate regularizers that improve the diversity of samples at the expense of their fidelity, resulting in more varied but less realistic samples as shown in the videos. Diffusion-based baselines suffer from *mode forcing* (see appendix D for more details) and generate spurious samples that are either excessively diverse or with little diversity. The prior best IMLE-based method (cIMLE) avoids spurious samples but falls short in image fidelity. In contrast, CHIMLE generates samples that are both diverse and of high fidelity. Figure 6 shows the qualitative comparison of the best reconstruction of the observed image found by different methods using IvO [61], which tries to approximately invert each model by optimizing over the latent code input to find a generated image closest to the observed image. As shown, the reconstruction of the CHIMLE model is the closest to the observed image and achieves almost pixel-level accuracy. In addition, we found that for our model, IvO can consistently converge to a high-quality sample, whereas it would diverge quite often for the baselines, suggesting that inverting our model is much easier than the baselines. This demonstrates that CHIMLE successfully covers the mode corresponding to the observed image.

## 3.3 Ablation Study

We incrementally remove (1) iterative combination (IC), (2) partial evaluation (PE), (3) latent code division (CD). As shown in Figure 7, each is critical to achieving the best results. Notably, we see an average increase in terms of FID by 7.7% after removing IC, by another 7.8% after removing PE and by another 12.8% after removing CD, thereby validating the importance of each of our contributions.

| | Night-to-day | | | Super-Resolution (SR) | | | Colourization (Col) | | | Image Decompression (DC) | | |
|---|---|---|---|---|---|---|---|---|---|---|---|---|
| | PRD ↑ | | IvO ↓ | PRD ↑ | | IvO ↓ | PRD ↑ | | IvO ↓ | PRD ↑ | | IvO ↓ |
| | $F_8$ (Recall) | $F_{1/8}$ (Precision) | | $F_8$ (Recall) | $F_{1/8}$ (Precision) | | $F_8$ (Recall) | $F_{1/8}$ (Precision) | | $F_8$ (Recall) | $F_{1/8}$ (Precision) | |
| *GAN-based:* | | | | | | | | | | | | |
| *BicycleGAN* [105] | 0.95 | **0.59** | 61.10 ± 0.23 | 0.90 | 0.39 | 15.26 ± 0.38 | 0.94 | 0.63 | 11.85 ± 1.78 | 0.94 | 0.71 | 9.70 ± 2.93 |
| *MSGAN* [60] | 0.96 | 0.50 | 86.3 ± 2.03 | 0.92 | 0.42 | 16.51 ± 0.64 | 0.93 | 0.62 | 11.60 ± 2.07 | **0.95** | 0.74 | 8.00 ± 1.10 |
| *DivCo* [56] | 0.94 | 0.46 | 59.50 ± 0.91 | 0.87 | 0.33 | 13.82 ± 0.32 | 0.92 | 0.61 | 10.88 ± 1.20 | 0.93 | 0.68 | 6.38 ± 1.08 |
| *MoNCE* [95] | 0.94 | 0.43 | 19.40 ± 0.01 | 0.85 | 0.35 | 56.23 ± 0.71 | 0.95 | 0.75 | 13.10 ± 0.79 | 0.92 | 0.76 | 23.50 ± 0.10 |
| *Diffusion-based:* | | | | | | | | | | | | |
| *DDRM* [42] | 0.38 | 0.12 | 134.00 ± 0.24 | 0.73 | 0.18 | 11.81 ± 0.05 | 0.80 | 0.44 | 13.70 ± 0.51 | 0.89 | 0.57 | 5.44 ± 0.01 |
| *NDM* [61] | 0.90 | 0.56 | – | 0.78 | 0.19 | – | **0.97** | 0.84 | – | 0.81 | 0.51 | – |
| *IMLE-based:* | | | | | | | | | | | | |
| *cIMLE* [53] | 0.96 | 0.58 | 1.54 ± 1.04 | 0.86 | 0.32 | 6.78 ± 0.43 | 0.92 | 0.64 | 0.71 ± 0.09 | 0.91 | 0.74 | 5.86 ± 0.20 |
| *CHIMLE (Ours)* | **0.97** | 0.58 | **0.96 ± 0.04** | **0.96** | **0.78** | **1.49 ± 0.11** | **0.97** | **0.87** | **0.32 ± 0.01** | **0.95** | **0.83** | **0.31 ± 0.10** |

Table 2: Comparison of mode accuracy, measured by $F_{1/8}$ score of the PRD metric [71], and mode coverage, measured by $F_8$ score of the PRD metric [71] and the inference-via-optimization metric (IvO) [61], between the model distribution and the true distribution by our method (CHIMLE) and the leading IMLE-based and general-purpose conditional image synthesis baselines. A higher value of the $F_8$ score shows better mode coverage (recall) and a higher value of the $F_{1/8}$ shows better mode accuracy (precision). A lower value of IvO shows better mode coverage since it indicate that the model can generate samples close to the observed images. For IvO, we show the average result across 5 independent runs on the scale of $10^{-3}$. As shown, CHIMLE outperforms or achieves comparable performance to the baselines.

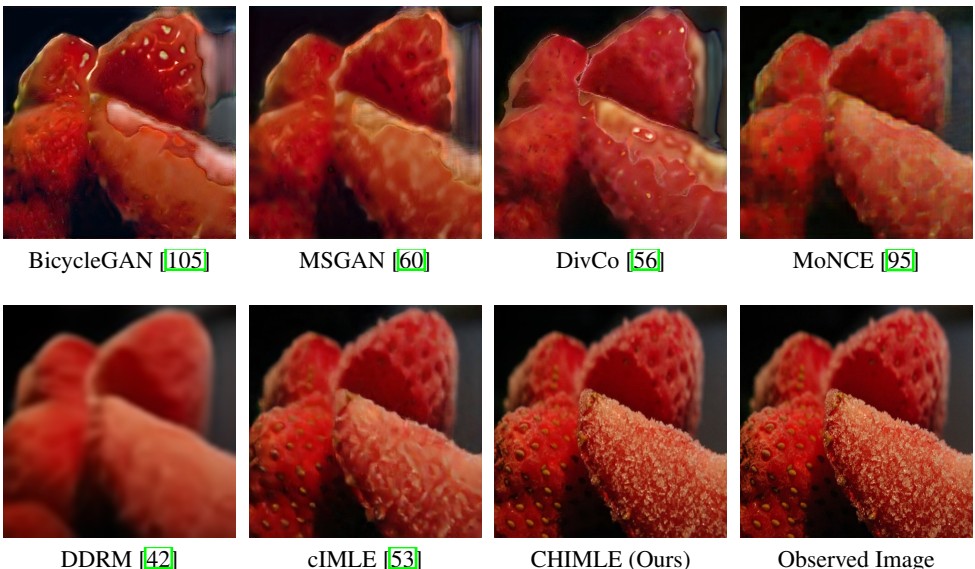

BicycleGAN [105]     MSGAN [60]     DivCo [56]     MoNCE [95]

DDRM [42]     cIMLE [53]     CHIMLE (Ours)     Observed Image

Figure 6: Qualitative comparison of best reconstructions of the observed image found by optimizing over the latent code input using the method of Inference via Optimization (IvO) [61]. The optimization starts by feeding a random latent code input to each method and minimizes the reconstruction loss from the model output to the observed image by optimizing the latent code input. As shown, the CHIMLE model allows for significantly better reconstruction of the observed image than the other methods, which demonstrates that the mode corresponding to the observed image is successfully covered by CHIMLE.

## 4 Related Work

**General-Purpose Conditional Image Synthesis**   Conditional image synthesis generate images from a conditioning input. The input can come in various forms, such as a class label [62, 64], a textual description [69], or an image [37]. The latter is sometimes known as image-to-image translation, which is the setting we consider. General image-conditional methods can be paired or unpaired. In the paired setting, each image in the source domain corresponds to an image in the target domain [37, 93, 86, 75]. In the unpaired setting, images in the source domain may not correspond to images in the target domain [104, 92, 81, 55, 16, 8, 67]. Whereas these previous methods are all unimodal, other methods aim for the more challenging task of multimodal synthesis in the paired setting [105, 24, 22] and in the unpaired setting [34, 50, 51, 17]. Since many of the aforementioned works use GANs as their generators which suffer from mode collapse, some works aim to address this issue by introducing mode seeking terms [60] or adding contrastive losses [56, 95, 33]. Another

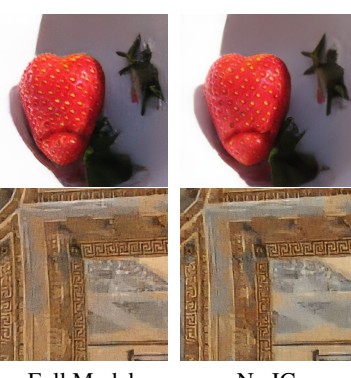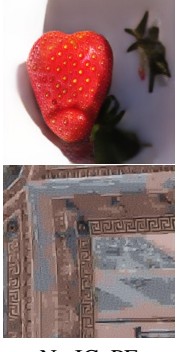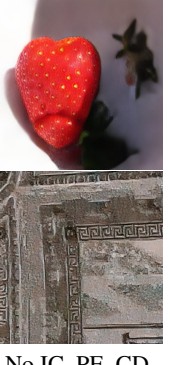

|  | SR | DC |
|---|---|---|
| Full Model | **16.01** | **73.69** |
| No IC | 17.06 | 81.24 |
| No IC, PE | 18.53 | 87.95 |
| No IC, PE, CD | 21.13 | 101.38 |

| Full Model | No IC | No IC, PE | No IC, PE, CD |
|---|---|---|---|

Figure 7: Qualitative (left figure) and quantitative (right table) of Fréchet Inception Distance (FID) comparison as we gradually remove (1) iterative combination (IC), (2) partial evaluation (PE), (3) latent code division (CD) on two tasks, $16\times$ super-resolution (SR) (first row in the figure) and image decompression (DC) (second row in the figure). As shown in the figure, the generated image contains less fine details (as in SR, where the strawberry seeds become less pronounced) or colour saturation (as in DC) as we remove each component, which is also reflected by the FID shown in the table.

line of work uses diffusion models [42, 6]. These methods can both generate spurious modes and drop modes (due to the problem of *mode forcing*, which we analyze in appendix D), and need to devote part of the model capacity to reconstruct the information encoded in the conditioning input from pure noise and retain it over the course of diffusion. In this paper, we focus on the paired image-conditional multimodal setting, and sidestep mode collapse by building on a recent generative modelling technique, conditional IMLE [53].

**Task-Specific Approaches** There is a large body of work on super-resolution, most of which consider upscaling factors of $2 - 4\times$. See [91, 63, 89] for comprehensive surveys. Many methods regress to the high-resolution image directly and differ widely in the architecture [23, 43, 47, 80, 28, 102, 19, 54]. Various conditional GANs have also been developed for the problem [49, 72, 88, 94, 66, 90, 87, 76]. For image colourization, a recent survey [3] provides a detailed overview of different methods. Some methods [14, 12, 100, 11] use deep networks to directly predict the colour and combine it with the input to produce the output. Other methods [36, 48, 21, 103, 79] take advantage of features from different levels or branches in the network for colour prediction. There is relatively little work on image decompression to our knowledge. [4] targets DCT-based compression methods and explicitly models the quantization errors of DCT coefficients. More work was done on learned image compression [2, 1], which changes the encoding of the compressed image itself. Another related area is image denoising, see [82] for a survey of deep learning methods. Many of the methods uses a ResNet [29] backbone or other variants of deep CNNs [97, 98, 99, 85].

## 5 Discussion and Conclusion

**Limitation** Our method can only generate output images that are similar to some training images. Take colourization as an example – if there were no green apple in the training data, then our model would not be able to generate green apples.

**Societal Impact** The TIM architecture adopts a modular design, its capacity could be expanded by adding more modules. While it may produce more refined results, training such a huge model may result in increased greenhouse gas emissions if the electricity used is generated from fossil fuels.

**Conclusion** In this paper, we developed an improved method for the challenging problem of multimodal conditional image synthesis based on IMLE. We addressed the undesirable tradeoff between sample efficiency and sample quality of the prior best IMLE-based method by introducing a novel hierarchical algorithm. We demonstrated on a wide range of tasks that the proposed method achieves the state-of-the-art results in image fidelity without compromising diversity or mode coverage.

**Acknowledgements** This research was supported by NSERC, WestGrid and the Digital Research Alliance of Canada. We thank Tristan Engst, Yanshu Zhang, Saurabh Mishra, Mehran Aghabozorgi and Shuman Peng for helpful comments and suggestions, and Kuan-Chieh Wang, Devin Guillory, Tianhao Zhang and Jason Lawrence for feedback on an early draft of this paper.

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
