# OpenReview forum: "CHIMLE: Conditional Hierarchical IMLE for Multimodal Conditional Image Synthesis"
_NeurIPS.cc/2022/Conference — NeurIPS 2022 Accept_

### Official Review · Reviewer_LstA · 2022-07-09

**Rating:** 6
**Confidence:** 4
**Soundness:** 3 good
**Presentation:** 2 fair
**Contribution:** 2 fair

**Summary:**

This paper presents a novel way of applying a non-adversarial image generation technique called IMLE to image-to-image problems.
Namely, the authors propose to split a latent code into several parts corresponding to different image scales. These sub-codes are selected coarse-to-fine: First, the code corresponding to the lowest resolution is found among the randomly sampled candidates. Afterward, this latent sub-code gets frozen, and subsequent parts of the code are chosen. As demonstrated in the paper, this hierarchical approach achieves the same quality as the baseline, with a lower number of generated samples.

According to the reported results of the evaluation, the presented approach outperforms several general-purpose image-to-image models, as well as some task-specific methods.

**Questions:**

I ask the authors to address the concerns listed above.

**Limitations:**

The authors have adequately addressed the limitations and potential negative societal impact of their work.

**Strengths And Weaknesses:**

1. Strengths.

    This paper proposes an interesting idea of incorporating multi-scale techniques into the IMLE-based generative model.
    This sound quite reasonable and it's a pleasure to see that this trick helps to reduce the number of required candidate samples, which is a common downside of IMLE.
     Moreover, the obtained results show that in the considered setting the presented solution outperforms even the task-specific baselines, which proves that CHIMLE may be useful for applications.
     Also, the authors have submitted the code, which is a good service for the community.

1. Weaknesses.

    * The explanation is sometimes unnecessarily wordy: while the method finally just employs the multi-scale search of latent codes, it is presented in a much more general way, although only one way of latent division is considered.
    * The evaluation uses F-measures computed from precision and recall. As shown in [1], these metrics struggle with issues, and it is more common to compute so-called Improved Precision and Recall [1].
    * Unfortunately, the resolution of generated images is not specified explicitly in the paper. Therefore, I am not sure if this approach may be scaled up to the image size of 1K. I ask the authors to provide the image size the model operates.
    * Why were not models with diverse outputs considered as task-specific baselines? E.g. for the case of super-resolution, SRFlow [2] may be a suitable one. This paper presents a novel way of applying a non-adversarial image generation technique called IMLE to image-to-image problems.


1. References

    [1] Kynkäänniemi et al. Improved Precision and Recall Metric for Assessing Generative Models. NeurIPS 2019.

    [2] Lugmayr et al. SRFlow: Learning the Super-Resolution Space with Normalizing Flow. ECCV 2020.

1. Post-rebuttal comments.

    I thank the authors for their feedback. It has addressed my main concerns. Therefore, I am inclined to increase the score.

---

> ### Author Response · Authors · 2022-08-02
> **Response to Reviewer LstA**
>
> ## Q1: The method is presented in a much more general way, although only one way is considered
>
> A1: This work considers the latent code search problem in the context of image generation, but the underlying principles (latent code division, partial evaluation, iterative combination) are more broadly applicable beyond image generation. We want to keep the description of the methodology general in order to help readers who might want to apply the method to other domains.
>
> ## Q2: Improved Precision and Recall Metric
>
> A2: As suggested, we computed the Improved Precision and Recall metric [a] and show the results compared to baselines in the table below.
>
> |            | Night-to-day        | Night-to-day     | SR                  | SR               | Col                 | Col              | DC                  | DC               |
> |------------|---------------------|------------------|---------------------|------------------|---------------------|------------------|---------------------|------------------|
> |            | Precision$\uparrow$ | Recall$\uparrow$ | Precision$\uparrow$ | Recall$\uparrow$ | Precision$\uparrow$ | Recall$\uparrow$ | Precision$\uparrow$ | Recall$\uparrow$ |
> | BicycleGAN | $0.522$             | $0.041$          | $0.615$             | $0.159$          | $0.744$             | $0.518$          | $\underline{0.869}$             | $\underline{0.486}$          |
> | MSGAN      | $0.479$             | $0.003$          | $0.545$             | $0.156$          | $0.694$             | $0.578$          | $0.766$             | $0.346$          |
> | DivCo      | $0.611$             | $0.007$          | $0.561$             | $0.153$          | $0.759$             | $0.484$          | $0.845$             | $0.310$          |
> | MoNCE      | $\textbf{0.818}$    | $0.008$          | $0.699$             | $0.120$          | $\textbf{0.787}$    | $\underline{0.624}$          | $0.830$             | $0.244$          |
> | cIMLE      | $0.578$             | $\underline{0.054}$          | $\underline{0.827}$             | $\underline{0.278}$          | $0.638$             | $0.423$          | $0.853$             | $0.441$          |
> | CHIMLE     | $\underline{0.785}$ | $\textbf{0.352}$ | $\textbf{0.934}$    | $\textbf{0.697}$ | $\underline{0.761}$ | $\textbf{0.757}$ | $\textbf{0.941}$    | $\textbf{0.717}$ |
>
> As shown in the table above, our proposed method outperforms all baselines by a significant margin across all tasks in recall, and in precision in most cases. In the few remaining cases, only one baseline outperforms our method, and it does so at the expense of a lower recall.
>
> ## Q3: What is the image size?
>
> A3: For Super-Resolution, our input is $32\times32$ and our output size is $512\times512$. For all other tasks, the input and target resolution are $256\times256$ and we downsample the input to the corresponding operating resolution at each level of the hierarchy. We will include this in the camera-ready. Regarding scaling up to image size of 1K, one can simply add an additional level in the hierarchy to reach that resolution.
>
> ## Q4: Comparison to SRFlow
>
> A4: As suggested, we started the training of SRFlow on super-resolution. ~~Although the training has not yet converged, we include the results we got so far and we will update them as the training progresses. The results table below shows the FID and faithfulness-weighted variance (FwV) achieved by SRFlow so far, alongside our results.~~
>
> [UPDATE]: The training has finished and we show the FID and faithfulness-weighted variance (FwV) results in the table below.
>
> |        | FID$\downarrow$  | FwV ($\sigma=0.2$)$\uparrow$ |
> |--------|------------------|------------------------------|
> | SRFlow |       $91.55$           |         $0.89$                     |
> | CHIMLE | $\textbf{16.01}$ | $\textbf{5.61}$              |
>
> ~~From the results shown in the table, so far our method is outperforming SRFlow.~~
>
> As shown in the table, our model outperforms SRFlow which validates the effectiveness of the proposed approach.
>
> [a] Kynkäänniemi et al. Improved Precision and Recall Metric for Assessing Generative Models. NeurIPS 2019.

---

> > ### Author Response · Authors · 2022-08-07
> > **Would appreciate any feedback from Reviewer LstA**
> >
> > We hope our response addressed your concerns. We would greatly appreciate any feedback and if you have any remaining concerns, we would be more than happy to clarify them before the discussion deadline on Aug 9th 8pm UTC / 4pm ET. Regarding the results on SRFlow, we are expecting to have the final results within the next 24hrs, thank you for your patience.

---

> > > ### Author Response · Authors · 2022-08-09
> > > **Update on Baseline Results**
> > >
> > > We have obtained the final results for SRFlow -- please refer to our original response for details. We would appreciate any feedback and are happy to clarify any further concerns if there are any.

---

### Official Review · Reviewer_VssD · 2022-07-09

**Rating:** 5
**Confidence:** 3
**Soundness:** 2 fair
**Presentation:** 3 good
**Contribution:** 3 good

**Summary:**

This paper aims to generate diverse and high-fidelity images in conditional image synthesis with a relatively small number of samples. Compared with the previous method conditional IMLE that counters the mode collapse, this paper improves the tradeoff between sampling efficiency and quality by introducing a diver-and-conquer selection mechanism by dimensions in the latent space. The proposed strategy brings better fidelity as well as the diversity of the generated results.

**Questions:**

1. As expressed in the weakness part, I have the following concern: does the hierarchical conditioned generation process deprecate the parallelism of this method?
2. And, how does the sampling efficiency connect to the proposed strategy that uses divide-and-conquer?
3. It is unclear whether the improved network architecture or the CHIMLE method contributes more to the SOTA performance. Can this be verified in new ablation studies that adopt the same generator architecture while using cIMLE, or adopt cIMLE's RRDB structure while using CHIMLE?

**Ethics Review Area:**

["I don’t know"]

**Limitations:**

The author has addressed the limitations and potential negative societal impact at the end of this paper.

**Strengths And Weaknesses:**

Strength:
+ The proposed divide-and-conquer strategy for latent code selection is very interesting. The division of latent code by dimensions is simple yet effective, and the partial evaluation of latent code components can well address the sub-problems with a modified criterion. When combing the latent code components, this method also considers the correlation between them and proposes the solve the sub-problems from low to high resolutions.
+ The intuition behind this proposed method is clearly explained.
+ The experimental results show great improvement on multiple tasks (16x super-resolution, night-to-day image translation, image colorization, extreme compression artifacts reduction)

Weakness:
- The divide-and-conquer algorithm aims to solve conceptually difficult problems with improved efficiency and parallelism. However, since this method requires constructing latent code at lower resolutions first, parallelism cannot be achieved. Besides, although the adoption of this time-tested algorithm into latent space sampling is very interesting, the author does not discuss too much about why high efficiency/low cost is vital for this problem. While the author does mention the sampling efficiency of CHIMLE is better than cIMLE, it is not very clear how it connects to the proposed strategy.
- This paper changes both network structure and the sampling strategy, while the experimental results do not include such ablation studies that disentangle the two. In the IMLE paper, the authors at least compare the SRIM and BicycleGAN with the same generator architecture to show the effectiveness of the proposed method, which could serve as a good example for this paper.

---

> ### Author Response · Authors · 2022-08-02
> **Response to Reviewer VssD**
>
> ## Q1: Does the hierarchical conditional generation process deprecate the parallelism of this method?
>
> A1: The proposed method is still parallelized over $m=100$ samples for each conditioning input —while it is not parallelized over different levels of the hierarchy, there are only $L=4$ of such levels which is much smaller than $m$.
>
> ## Q2: How does sampling efficiency connect to the proposed strategy?
>
> A2: The proposed divide-and-conquer strategy reduces the search space for the latent code to a more promising region. Because the region is smaller, there are more samples generated within a given area within the region than outside of it. This makes it more likely to find a sample that is close to the observed image, which leads to better sampling efficiency.
>
> ## Q3: Whether the improved architecture or the method contributes more to the SOTA performance?
>
> A3: We performed the suggested ablation study and trained cIMLE using the same architecture our method uses on two tasks (Super-resolution and Colourization) to disentangle the effect of the sampling strategy and network architecture. We find that our method still outperforms cIMLE by 33.6% on average with the same network architecture, which validates the effectiveness of our method.
>
> In addition, we retrained various GAN-based baselines (BicycleGAN, MSGAN and MoNCE) with our architecture to further validate our method’s effectiveness. We observed that the GAN-based baselines failed to converge when trained from scratch with our architecture, so we pretrained their generator using our method which gave them an advantage. We show the FID results in the table below.
>
> |                               | Super-Resolution (SR) | Colourization (Col) |
> |-------------------------------|-----------------------|---------------------|
> | BicycleGAN + our architecture | $53.30$               | $66.32$             |
> | MSGAN + our architecture      | $57.94$               | $81.86$             |
> | MoNCE + our architecture      | $31.72$               | $\underline{27.85}$             |
> | cIMLE + our architecture      | $\underline{21.13}$               | $42.67$             |
> | CHIMLE                        | $\textbf{16.01}$      | $\textbf{24.33}$    |
>
> As shown above, our method consistently outperforms the baselines which demonstrates the effectiveness of our method.

---

> > ### Author Response · Authors · 2022-08-07
> > **Would appreciate any feedback from Reviewer VssD**
> >
> > We hope our response addressed your concerns. We would greatly appreciate any feedback and if you have any remaining concerns, we would be more than happy to clarify them before the discussion deadline on Aug 9th 8pm UTC / 4pm ET.

---

> > ### Comment · Reviewer_VssD · 2022-08-09
> > **The authors addressed some of my questions**
> >
> > For Q1: the hierarchical conditioned generation process does deprecate the parallelism of inference within the model.
> >
> > For Q2: the author still does not discuss too much about why high efficiency/low cost is vital for this problem (as described in weakness).
> >
> > For Q3: the author compared with multiple methods with the same architecture to show the effectiveness of proposed strategy, which solves my question.
> >
> > The author has solved some of my questions, while missing to answer Q2 well.

---

> > > ### Author Response · Authors · 2022-08-09
> > > **Clarifications for your questions**
> > >
> > > **For Q1**: We want to remind the reviewer that while the generation procedure is hierarchical, the number of such levels of hierarchy is a lot smaller than the number of parallel generated samples, so it does not compromise generation speed. On the contrary, the hierarchical generation procedure greatly improves the generation efficiency, as shown in Figure 2.
> > >
> > > **For Q2**: As discussed on lines 85-89 in Section 2.2, achieving high generated image quality requires generating many samples. However, sample generation is expensive and this limits the performance of cIMLE in practice. The proposed method overcomes this limitation by generating samples more efficiently. The results show that this is critical to improving generated image quality — our method substantially improves in FID by 33.6% on average compared to “cIMLE + our architecture” (which only differs in the sampling procedure from our method).
> > >
> > > Hope this helps clarify your questions; please let us know if you have any further comments or questions.

---

### Official Review · Reviewer_WPok · 2022-07-11

**Rating:** 5
**Confidence:** 4
**Soundness:** 3 good
**Presentation:** 3 good
**Contribution:** 2 fair

**Summary:**

The paper focuses on "one-shot" (so the SOTA autoregressive transformer and diffusion models are not considered for the comparison in this paper) multi-modal conditional generative tasks. The proposed method is based on the IMLE algorithm. The authors reduce the computational cost (in the training stage) of the previous conditional IMLE framework by a divide and conquer strategy. Experiments are conducted on night-to-day, monocular image super-resolution, image colorization, image decompression, and layout-to-image tasks to validate the effectiveness of the proposed method over the previous conditional IMLE scheme.

**Questions:**

Please see Weakness.

**Limitations:**

The authors address the limitations and potential social impact.

**Strengths And Weaknesses:**

*Strengths*
1. The paper is well-written, and the idea of divide and conquer for improving cIMLE is sound.
2. The authors provide a clear introduction to the previous cIMLE method, which is very helpful for readers unfamiliar with this field.
3. According to extensive experimental results, the proposed method synthesizes more realistic and diverse images compared to several baseline approaches.
*Weakness*
1. Several metrics are proposed to evaluate the realism and diversity of the generated images, e.g., precision and recall. Can the authors provide more intuitions why these metrics are not used in measuring the diversity in conditional generative modeling?
- Kynkäänniemi et al., "Improved Precision and Recall Metric for Assessing Generative Models."
2. Recent data augmentations can significantly improve the performance of GAN-based approaches, e.g., ADA, especially in the limited training data case. Can the authors comment on 1) how the proposed IMLE-based performs against the GAN-based methods trained with ADA and 2) how the proposed algorithm performs in the limited training data case?
- Karras et al., "Training Generative Adversarial Networks with Limited Data."

---

> ### Author Response · Authors · 2022-08-02
> **Response to Reviewer WPok**
>
> ## Q1: Improved Precision and Recall Metric
>
> A1: As suggested, we computed the Improved Precision and Recall metric [a] and show the results compared to baselines in the table below.
>
> |            | Night-to-day        | Night-to-day     | SR                  | SR               | Col                 | Col              | DC                  | DC               |
> |------------|---------------------|------------------|---------------------|------------------|---------------------|------------------|---------------------|------------------|
> |            | Precision$\uparrow$ | Recall$\uparrow$ | Precision$\uparrow$ | Recall$\uparrow$ | Precision$\uparrow$ | Recall$\uparrow$ | Precision$\uparrow$ | Recall$\uparrow$ |
> | BicycleGAN | $0.522$             | $0.041$          | $0.615$             | $0.159$          | $0.744$             | $0.518$          | $\underline{0.869}$             | $\underline{0.486}$          |
> | MSGAN      | $0.479$             | $0.003$          | $0.545$             | $0.156$          | $0.694$             | $0.578$          | $0.766$             | $0.346$          |
> | DivCo      | $0.611$             | $0.007$          | $0.561$             | $0.153$          | $0.759$             | $0.484$          | $0.845$             | $0.310$          |
> | MoNCE      | $\textbf{0.818}$    | $0.008$          | $0.699$             | $0.120$          | $\textbf{0.787}$    | $\underline{0.624}$          | $0.830$             | $0.244$          |
> | cIMLE      | $0.578$             | $\underline{0.054}$          | $\underline{0.827}$             | $\underline{0.278}$          | $0.638$             | $0.423$          | $0.853$             | $0.441$          |
> | CHIMLE     | $\underline{0.785}$ | $\textbf{0.352}$ | $\textbf{0.934}$    | $\textbf{0.697}$ | $\underline{0.761}$ | $\textbf{0.757}$ | $\textbf{0.941}$    | $\textbf{0.717}$ |
>
> As shown in the table above, our proposed method outperforms all baselines by a significant margin across all tasks in recall, and in precision in most cases. In the few remaining cases, only one baseline outperforms our method, and it does so at the expense of a lower recall.
>
> ## Q2: GAN-based Method Trained with ADA
>
> A2: As suggested, we compared to GAN trained with ADA in the limited data setting. We choose the task with fewest training images (980 images), image decompression, for the comparison under this setting. We choose the best performing GAN-based baseline for that task, BicycleGAN, and trained it with ADA. The FID result is shown in the table below.
>
> |                  | FID$\downarrow$  |
> |------------------|------------------|
> | BicycleGAN       | $87.35$          |
> | BicycleGAN + ADA | $\underline{86.55}$          |
> | CHIMLE           | $\textbf{73.69}$ |
>
> As shown in the table above, our proposed method outperforms BicycleGAN trained with ADA. This result also demonstrates that our proposed method performs well in the limited training data case.
>
> ## Reference
>
> [a] Kynkäänniemi et al. Improved Precision and Recall Metric for Assessing Generative Models. NeurIPS 2019.

---

> > ### Author Response · Authors · 2022-08-07
> > **Would appreciate any feedback from Reviewer WPok**
> >
> > We hope our response addressed your concerns. We would greatly appreciate any feedback and if you have any remaining concerns, we would be more than happy to clarify them before the discussion deadline on Aug 9th 8pm UTC / 4pm ET.

---

> > > ### Comment · Reviewer_WPok · 2022-08-08
> > > **Feedback to authors' response**
> > >
> > > Thank you for the responses. I think these address my concern in raised in the review.

---

### Official Review · Reviewer_UYaB · 2022-07-11

**Rating:** 5
**Confidence:** 3
**Soundness:** 2 fair
**Presentation:** 2 fair
**Contribution:** 2 fair

**Summary:**

This work proposes a new method Conditional Hierarchical IMLE to get around limitation of requirements of a large number of samples to generate high-fidelity images.
The proposed CHIMLE is shown to improve generated image fidelity, with a clear reduction in Fréchet Inception Distance compared to the prior best IMLE-based method.

**Questions:**

See above comments.

**Limitations:**

The limitations and  social impact have been discussed in the manuscript.

**Strengths And Weaknesses:**


Strengths:
To generate samples in a way such that the best sample is about as similar to the observed image as if a large number of samples had been generated without actually generating that many samples,
the author proposes several methods, including partitioning the latent code, partial evaluation of latent code components, and iterative construction of latent code. These ideas give rise to a the method of Conditional Hierarchical IMLE or CHIMLE.
This work demonstrates that CHIMLE significantly outperforms the prior best IMLE-based method in terms of both fidelity and diversity on a variety of tasks. Besides, they also show that CHIMLE achieves superior image fidelity and mode coverage compared to leading general-purpose multimodal and task-specific methods.



Weaknesses:
1. In the experiments, what backbones are adopted in CHIMLE for different tasks, e.g., colorization, super resolution. If the comparison fair as different methods use different generator structure, e.g., BicycleGAN, MSGAN.
2. Some proper metrics should be used for different tasks, e.g., PSNR for super-resolution.
3. Please check the citation forms, I find some citations are in false format, e.g.:
Yingchen Yu Rongliang Wu Shijian Lu Fangneng Zhan, Jiahui Zhang. Modulated contrast for versatile image synthesis. In Proceedings of the IEEE/CVF Conference on Computer Vision and Pattern Recognition, 2022.

---

> ### Author Response · Authors · 2022-08-02
> **Response to Reviewer UYaB**
>
> ## Q1: Baseline Comparison with the Same Backbone Architecture
>
> A1: As suggested, we retrained BicycleGAN and MSGAN on super-resolution (SR) and colourization (Col) using the same generator architecture used by our method. Furthermore, we also retrained two other baselines, cIMLE and MoNCE, with the same architecture. We observed that the GAN-based baselines failed to converge when trained from scratch with our architecture, so we pretrained their generator using our method (which gave them an advantage over the vanilla randomly initialized versions). We show the FID results in the table below.
>
> |            | Super-Resolution (SR) | Colourization (Col) |
> |------------|-----------------------|---------------------|
> | BicycleGAN + our architecture | $53.30$               | $66.32$             |
> | MSGAN + our architecture      | $57.94$               | $81.86$             |
> | MoNCE + our architecture      | $31.72$               | $\underline{27.85}$             |
> | cIMLE + our architecture      | $\underline{21.13}$               | $42.67$             |
> | CHIMLE     | $\textbf{16.01}$      | $\textbf{24.33}$    |
>
> As shown above, our method still consistently outperforms the baselines with the same network architecture, thereby validating the effectiveness of our method.
>
> ## Q2: What is the backbone architecture?
>
> A2: Please refer to Section 3 of the paper and Section A of the supplementary materials for details.
>
> ## Q3: PSNR for Super-Resolution (SR)
>
> A3: As suggested, we computed the PSNR metric for our method and the baseline methods on SR and show the results in the table below.
>
> |                               | PSNR$\uparrow$   |
> |-------------------------------|------------------|
> | BicycleGAN                    | $15.99$          |
> | BicycleGAN + our architecture | $17.83$          |
> | MSGAN                         | $16.18$          |
> | MSGAN + our architecture      | $15.67$          |
> | MoNCE                         | $18.47$          |
> | MoNCE + our architecture      | $19.42$          |
> | RFB-ESRGAN                    | $\underline{20.13}$          |
> | cIMLE                         | $20.11$          |
> | cIMLE + our architecture      | $19.97$          |
> | CHIMLE                        | $\textbf{20.30}$ |
>
> As shown above, our method outperforms the baselines on SR in terms of PSNR, which validates our method’s performance.
>
> ## Q4: Citation Format
>
> A4: Good catch, we will fix it in the camera-ready.

---

> > ### Author Response · Authors · 2022-08-07
> > **Would appreciate any feedback from Reviewer UYaB**
> >
> > We hope our response addressed your concerns. We would greatly appreciate any feedback and if you have any remaining concerns, we would be more than happy to clarify them before the discussion deadline on Aug 9th 8pm UTC / 4pm ET.

---

### Author Response · Authors · 2022-08-02
**General Response**

We thank all reviewers for your time, constructive comments and unanimous appreciation of the proposed method and the good results. In particular, the reviewers remarked “the proposed divide-and-conquer strategy […] is very interesting” (R VssD), “the paper is well-written, and the idea […] is sound”(R WPok), “this paper proposes an interesting idea […] sounds quite reasonable” (R LstA), “this work demonstrates that CHIMLE significantly outperforms the prior best IMLE-based method […] achieves superior image fidelity and mode coverage” (R UYaB) and “the results show great improvement on multiple tasks” (R VssD).

We found the comments and questions very helpful, which add value to our work. Here, we provide a one-sentence summary of our response to questions raised by the reviewers — please refer to our individual responses to each review for the details.

### Q1: What is the effect of using the proposed architecture for the baselines? (R UYaB, R VssD)

A1: We have tried this, and found that our method still consistently outperformed the baselines.


### Q2: Improved Precision and Recall Metric [a]? (R WPok, R LstA)

A2: Our method achieves the best or nearly the best performance across tasks.

### Q3: PSNR for Super-Resolution (SR)? (R UYaB)

A3: As suggested, we computed the PSNR metric on super-resolution and found our method outperformed the baselines.

### Q4: Comparison to GAN trained with ADA on limited training data (R WPok)

A4: We tried this, and found that our method outperformed the GAN-based baseline under this setting.

### Q5: Does the hierarchical conditioned generation process deprecate parallelism of this method? (R VssD)

A5: No, the method parallelizes over the generation of different samples.

### Q6: The method is presented in a general way, although only one way is considered (R LstA)

A6: We presented it in a general way to make it easier for readers who might want to apply the underlying ideas to contexts beyond image generation.

### Q7: Not sure whether this approach may be scaled up to image size of 1K? (R LstA)

A7: Yes, it can, by adding one more level to the hierarchy.

### Reference
[a] Kynkäänniemi et al. Improved Precision and Recall Metric for Assessing Generative Models. NeurIPS 2019.

---

### Meta-Review · Area_Chair_h6am · 2022-08-28

**Recommendation:** Accept
**Confidence:** Certain

**Metareview:**

This paper introduces a conditional image synthesis method based on Implicit Maximum Likelihood Estimation (IMLE). Compared to previous work CIMLE, the paper has introduced a divide-and-conquer method to accurately estimate latent code without evaluating many samples. The paper has received consistently positive reviews. Reviewers found the idea intuitive and interesting, and the method effective (especially compared to CIMLE). The rebuttal further addressed the concerns and included comparisons with the same backbone architecture and additional baselines and evaluations. The AC agreed with the reviewers’ consensus and recommended accepting the paper.


**Award:**

No

---

### Decision · Program_Chairs · 2022-09-14

Accept